# Performance Degradation in Static Random Access Memory of 10 nm Node FinFET Owing to Displacement Defects

**DOI:** 10.3390/mi14051090

**Published:** 2023-05-22

**Authors:** Minji Bang, Jonghyeon Ha, Gyeongyeop Lee, Minki Suh, Jungsik Kim

**Affiliations:** Department of Electrical Engineering, Gyeongsang National University, Jinju 52828, Gyeongnam, Republic of Korea; minjibang@gnu.ac.kr (M.B.); jh990502@gnu.ac.kr (J.H.); rudduq1660@gnu.ac.kr (G.L.); alsrl8588@gnu.ac.kr (M.S.)

**Keywords:** displacement defect, fin field-effect-transistor, cosmic rays, terrestrial radiation, technology computer-aided design (TCAD)

## Abstract

We comprehensively investigate displacement-defect-induced current and static noise margin variations in six-transistor (6T) static random access memory (SRAM) based on a 10 nm node fin field-effect transistor (FinFET) using technology computer-aided design (TCAD). Various defect cluster conditions and fin structures are considered as variables to estimate the worst-case scenario for displacement defects. The rectangular defect clusters capture more widely distributed charges at the fin top, reducing the on- and off-current. The read static noise margin (RSNM) is the most degraded in the pull-down transistor during the read operation. The increased fin width decreases the RSNM due to the gate field. The current per cross-sectional area increases when the fin height decreases, but the energy barrier lowering by the gate field is similar. Therefore, the reduced fin width and increased fin height structure suit the 10 nm node FinFET 6T SRAMs with high radiation hardness.

## 1. Introduction

Static random access memory (SRAM) is used as CPU cache memory, occupying over half of the System-on-Chip (SoC) die area. The high density of SRAM with nanosized transistors is vulnerable to variability due to process variations and short-channel effects such as drain-induced barrier lowering (DIBL), non-ideal subthreshold swing (*SS*), and threshold voltage (V_th_) roll-off [1,2,3,4]. The fin field-effect transistor (FinFET)-based SRAM has been adopted to overcome the problems of planar SRAM [5]. The tri-gate structure of the FinFET provides high gate control in the channel region to suppress short-channel effects. FinFET 6T SRAMs are used in high-speed and high-reliability applications such as automotive and aerospace systems. They are also ideal for low-power portable devices such as smartphones, tablets, and wearables due to their low power consumption and small size. However, an aggressively scaled down technology increases the radiation sensitivity of the device. Below 22 nm, the defect and channel volumes become similar; therefore, the effect of radiation cannot be ignored [6].

Radiation effects in semiconductor devices have been classified into the total ionizing dose, single event upset, and displacement defects. Total ionizing dose effects are caused by the accumulation of ionizing radiation over time when an electronic device is exposed to high-energy particles such as protons and gamma rays. The ionizing radiation effects are caused by interface and oxide traps by forming electron-hole pairs at the SiO_2_ and Si/SiO_2_ interfaces. The accumulation of charge at the interface and oxide traps modifies the threshold voltage of the silicon device, causing soft errors. Total ionizing dose effects also affect device performance by changing the mobility of charge carriers [7,8]. Single event upset occurs when high-energy radiation particles, such as neutrons and protons, collide with a memory cell and the collected charge is greater than the critical charge required to cause an error. When the incident particles collide with the memory cell, they transfer energy to the silicon atoms, creating electron-hole pairs that induce a transient current pulse. The transient current pulse potentially causes a state change in the memory cell and results in data errors [7,9]. Displacement defects occur when high-energy radiation particles, such as neutrons, strike lattice atoms, causing them to dislodge from the normal lattice. Lattice defects are created when the initially collided and dislodged atom, the primary knock-on atom (PKA), transfers its kinetic energy to an adjacent atom. The displaced atoms form vacancies where atoms are dislodged from their lattice positions and interstitials where the dislodged atoms are in non-lattice positions. Divacancies, resulting from the formation of two adjacent vacancies, represent a prevalent type of defect observed in semiconductors. The formation of point defects and defect clusters or solely isolated defects is influenced by several factors, including the energy of the incident radiation particles [10,11,12]. Terrestrial neutron energies span from millielectronvolts to gigaelectronvolts, with the most prevalent range being 1–100 MeV. Fast neutrons with energies greater than 1 MeV can cause significant lattice defects due to the silicon threshold displacement energy of 50 keV [13]. 

The event rate of fast neutron scattering on silicon atoms is calculated using the product of the neutron flux (φ, neutrons/cm^2^ ⋅ h), number of target atoms (*N*, atoms/cm^3^), and cross-section (σ, 1 barn = 10^−24^ cm^2^). The flux and spectrum of fast neutrons vary with temperature, latitude, and longitude. To observe the event rate, experiments were conducted at the maximum flux of 5 × 10^13^/cm^2^·s at the AP-1000 reactor neutron irradiation facility in the United States [14]. The number of silicon atoms per unit volume was 5 × 10^22^/cm^3^, and the cross-section of a neutron at 1 MeV was 2 barns (2 × 10^−24^ cm^2^). With these values, it was calculated that in a silicon area of 1 cm^3^, 5 × 10^12^/cm^3^⋅ s events occur. Therefore, in a volume of 0.0155 μm^3^ (i.e., a cross-sectional area of 0.031 μm^2^ × 0.5 μm) of FinFET 6T SRAMs, irradiated for 700 s, 54 events occur within the defect cluster.

Comparing this to the situation at the New York sea level, it takes more than 100 years for 54 displacement defect events to occur in a single cell of FinFET 6T SRAMs [15]. However, for an integrated circuit consisting of many cells, an event occurs approximately every 21 days. The results obtained from experiments performed on the fast neutron scattering on silicon atoms suggest that radiation exposure can cause a significant number of events to occur in FinFET 6T SRAMs, resulting in malfunction and permanent damage. These experimental results emphasize the need to ensure the high reliability of devices against radiation, especially in radiation-exposed environments such as space exploration or nuclear power plants. Despite the significance of displacement defects, radiation studies have been limited to soft errors, such as total ionizing dose and single event upset [16,17,18]. Therefore, the effect of displacement defects on the FinFET 6T SRAMs should be investigated.

In this work, we thoroughly investigate the impact of displacement defects on the DC characteristics of FinFET 6T SRAMs. We analyze the on- and off-current for defect clusters with two cross-sectional shapes. The static noise margin (SNM) variation due to displacement defects is observed under different SRAM operations. We propose a method to mitigate the effect of displacement defects through variations in the fin structure. Finally, we provide further studies on the number of defects and the locations of defect clusters along fin heights.

## 2. Simulation Modeling Methodology

In this work, 10 nm node FinFET 6T SRAMs were simulated using TCAD simulation [19]. A high-field saturation model was adopted with the inversion and accumulation layer mobility model to consider the carrier mobility owing to the strong electric field in the short channel [20,21]. The Shockley-Read-Hall recombination model, Hurkx band-to-band tunneling model, and Auger recombination were used to consider recombination. The modified local-density approximation (MLDA) model considered quantum confinement effects at the interface between semiconductors and insulators [22]. To increase hole mobility, the Piezo model was adopted to consider the stress between SiGe and Si in the PMOS field-effect transistor (FET) [23,24,25]. The Philips unified mobility was used to account for carrier-carrier scattering and impurity scattering.

Figure 1a shows the 3D schematic of the FinFET 6T SRAMs. Figure 1b shows the calibrated *I_d_*–*V_g_* transfer characteristics, which represent the main electrical characteristics of the FinFET 6T SRAMs, using data from [26]. Table 1 defines the geometry and doping concentration of the FinFET 6T SRAMs. The gate length of the tapered FinFET is 20 nm, and the fin width and fin height are 8 nm and 46 nm, respectively. The specific tapered fin width is as follows:8 nm at the top of the fin (fin height = 2 nm);9 nm at the middle of the fin (fin height = 20 nm);11 nm at the bottom of the fin (fin height = 40 nm).

The contact poly pitch (CPP) was 54 nm, and the effective oxide thickness (EOT) was 1.4 nm. The source/drain (S/D) and channel doping concentrations were 1 × 10^20^ and 5 × 10^16^ cm^−3^, respectively.

Radiation was randomly located in the semiconductor, and displacement defects mainly degraded devices in the regions where the current flowed [27]. Various displacement defect positions were considered to identify the worst location of the defects in the FinFET using TCAD simulation [6]. The defect cluster was located at the top of the fin height (X-direction) and the center of the gate length (Y-direction) and fin width (Z-direction) because the applied voltage of the S/D depended on the read and write operation. To reproduce displacement defects, we created 54 defects in a rectangular cluster [28,29]. Displacement defects were modeled at various energies: the deep trap energy level (acceptor-like trap, *E_c_* − 0.4 eV) for the pass gate (PG) and pull-down (PD) NMOS FET and the shallow trap energy level (donor-like trap, *E_v_* + 0.2 eV) for the pull-up (PU) PMOS FET [6]. The on-state current was extracted at *V_g_* = *V_d_* = 0.7 V, and the off-state current was extracted at *V_g_* = 0 V and *V_d_* = 0.7 V. The degradation rate was defined as (pre-irradiated value − displacement defects-induced value)/(pre-irradiated value) × 100 (%).

## 3. Results and Discussion

For the on-state current density, Figure 2a shows two defect clusters with 54 defects: the rectangular defect cluster 6 × 3.75 × 2 nm^3^ and square defect cluster 3 × 3.75 × 3 nm^3^. Figure 2b shows the off-state current density of the fin. The current variation depends on the cross-sectional area where the defect cluster occupies the current density, as shown in Figure 2c. The rectangular defect cluster further reduces the current compared to the square defect cluster, and the on-current reduction due to displacement defects is large in the PD1 NMOS FET, up to 28.95% of the pre-irradiation value. Because of the large area that occupies the current density in the rectangular defect cluster, many carriers are captured, further reducing the carrier mobility and current density.

The deep trap energy level captures electrons, causing the raised channel’s energy barrier to interfere with the current flow. Displacement defects act as the trapping center that increases the leakage current through tunneling in the channel region [30]. However, the undoped channel and lightly doped drain (LDD) minimize the trap-assisted tunneling (TAT) mechanism, reducing the off-current due to displacement defects. The acceptor-like trap of the deep trap energy level is close to the mid-band gap, requiring more energy to release electrons. On the other hand, the donor-like trap of the shallow trap energy level requires less energy to emit the holes. When *V_g_* = 0 V and *V_d_* = 0.7 V, the shallow trap energy level is below the fermi level, and there are fewer carriers to capture. This means that the off-current degradation rate of the NMOS FET is higher than that of the PMOS FET.

Figure 3a shows the variation in the SNM due to displacement defects during the read and write operation. The SNM was the maximum amount of DC noise applied to an input without reversing the output state. Sufficiently large DC noise did not guarantee the stable operation of the SRAM [31].

Displacement defects were generated in the single transistor (PG1 NMOS FET, PD1 NMOS FET, and PU1 PMOS FET) and coupled transistors (PG1 + PG2 NMOS FET, PD1 + PD2 NMOS FET, and PU1 + PU2 PMOS FET) located in the right inverter. The drive strength of the transistor for the read and write operation determines the worst transistor due to displacement defects. Table 2 shows the read static noise margin (RSNM) and write static noise margin (WSNM) values due to displacement defects.

In the write operation, the PG1 NMOS FET is most sensitive to displacement defects, reducing the WSNM by up to 2.1 mV. Figure 3b shows that bit-line (BL) = 0 V and bit-line bar (BLB) = 0.7 V were applied when the word-line (WL) was selected. Data “0” on the input voltage (*V_in_*) node and data “1” on the output voltage (*V_out_*) node were stored. The write current flows from *V_dd_* to the BLB through the PG1 NMOS FET. The discharging strength of the PG1 NMOS FET should be higher than the restoring strength of the PU1 PMOS FET. Displacement defects induced in the PG1 NMOS FET reduce the *gamma ratio* between the drive current of the PG1 NMOS FET and that of the PU1 PMOS FET, reducing the write ability. However, the write operation is not sensitive to slight voltage fluctuations to write other data by changing the storage node’s data state (*V_in_*, *V_out_*). Therefore, due to displacement defects, the WSNM has a less than 1% degradation rate for the pre-irradiation value.

On the other hand, the read operation is highly affected by displacement defects because it is sensitive to slight voltage fluctuations to read data from storage nodes using a sense amplifier [31]. The RSNM is reduced by up to 35.87 mV when displacement defects are induced in the PD1 NMOS FET. Figure 3c illustrates that both BL and BLB were pre-charged with 0.7 V, and when WL was selected, data “0” and “1” were stored in the *V_in_* and *V_out_* nodes, respectively. The PG1 NMOS FET is turned on, and the read current flows through the PD1 NMOS FET to the ground. Displacement defects induced in the PD1 NMOS FET reduce the driving strength, reducing the *beta ratio* between the drive current of the PD1 NMOS FET and that of the PG1 NMOS FET.

Figure 4a,b show the left and right inverter flips. The left voltage transfer curve (left VTC, Flip L) of Figure 4c indicates that the output voltage (*V_out_*) of the right inverter is flipped. The right VTC (Flip R) is the input voltage (*V_in_*) flipped from the left inverter. The butterfly curve shows the RSNM of the PD1 NMOS FET and PD1 + PD2 NMOS FET damaged by displacement defects, as shown in Figure 4c. The RSNM was defined as the maximum size of a rectangle entering the closed area of the curve [31]. The RSNM was extracted from the right inverter to analyze the effect of displacement defects in a single transistor and coupled transistors.

Interestingly, the largest RSNM reduction due to displacement defects was in the PD1 NMOS FET condition. In the PD1 NMOS FET and PD1 + PD2 NMOS FET induced by displacement defects, the RSNM is reduced to 39.40 mV and 22.68 mV, respectively. The V_th_ shift owing to displacement defects increases the right VTC in the vertical direction at the region above *V_in_* = 0.35 V. Displacement defects do not damage the left inverter at the region below *V_out_* = 0.35 V; therefore, the left VTC is fixed without shifting. The closed area of the butterfly curve decreases when the right VTC increases in the vertical direction at the region above *V_in_* = 0.35 V and below *V_out_* = 0.35 V. The butterfly curve intersection point of the PD1 NMOS FET condition deviates from the intersection point of the pre-irradiation, resulting in a mismatch.

The reduction in the RSNM in the PD1 + PD2 NMOS FET condition due to displacement defects is 16.72 mV lower than that of the PD1 NMOS FET. Unlike the PD1 NMOS FET condition, the left and right VTCs shift under the PD1 + PD2 NMOS FET condition. The closed area is relatively larger than the PD1 NMOS FET condition, reducing the mismatch on the slope of the pre-irradiation.

Figure 5a,b show the effect of the fin structure on the current variations due to displacement defects. Three fin structures are selected to observe the worst condition. The fin structural variations are (I) fin width = 6 nm, fin height = 46 nm; (II) fin width = 8 nm, fin height = 46 nm; and (III) fin width = 8 nm, fin height = 38 nm. The short-channel effects analysis is omitted due to the consistent gate lengths under variable conditions of fin structure.

Under conditions (I) and (II), the current per cross-sectional area is 5.87 × 10^6^ and 4.76 × 10^6^ A/cm^2^, respectively. When the fin width is reduced, the gate field sufficiently suppresses the energy barrier of the channel raised by displacement defects, as shown in Figure 5c. Therefore, the current is reduced less in (I) because the rectangular defect cluster of (I) is less able to capture charge carriers than in (II).

Figure 5b shows the on- and off-current decrease by 31.20% and 32.18% in (III). The current per cross-sectional area in (III) increases to 4.93 × 10^6^ A/cm^2^, resulting in more charges trapped than in (II). Unlike when the fin width is increased, the energy level of the conduction band is lowered by the suppression of the gate field in (III), which is 3.10% higher than in (II). The effect of the trapped charge is superior to that of the gate field.

Figure 5d shows that the displacement defects in (III) most degrade the RSNM. The RSNMs of 39.4 and 22.68 mV are reduced in the PD1 NMOS FET and PD1 + PD2 NMOS FET, from which displacement defects are induced. This indicates a direct correlation between the decrease in the on- and off-current and the RSNM.

For the worst-case scenario due to displacement defects, we conducted studies with the fin width and height of 8 and 38 nm, respectively. Figure 6a,b compare the current and RSNM for the variation in the number of defects in the rectangular defect cluster located in different regions along the fin height of the PD1 NMOS FET. The presence of defect clusters at the top of the fin interferes with the main current path, resulting in a reduction of up to 31.61% in the on-current compared to the middle of the fin. Similarly, the off-current at the middle of the fin is reduced by up to 53.17%. The top of the fin exhibits less leakage current due to the improved current flow control offered by the tri-gate. Conversely, the middle of the fin, which is influenced by the double-gate, has a higher off-current. The degradation of the RSNM due to displacement defects is predominantly influenced by the on-current, thereby rendering the defect clusters generated at the top of the fin detrimental to the RSNM by up to 32.93%.

Moreover, it observed that the degradation of both the current and RSNM caused by displacement defects reaches a saturation point with an increase in the number of defects within the cluster. The saturation effect is attributed to the fact that as the number of carriers captured from the trapping center by the defect cluster increases, it eventually converges to the number of inversion carriers. This phenomenon is due to the non-linear behavior of the energy barrier caused by displacement defects, which does not increase proportionally with the number of defects. Consequently, the linear reduction in the carrier mobility caused by displacement defects no longer persists beyond a certain point, resulting in a saturated degradation rate.

## 4. Conclusions

In this study, we analyze the effect of displacement defects on FinFET 6T SRAMs using TCAD simulation. Unlike the square defect cluster, the rectangular defect cluster with a large cross-sectional area where the defect cluster occupies the current density has a reduced on-current. Displacement defects induce different influences depending on the operation characteristics of the SRAM. Displacement defects are ignored in the write operation because they reduce the WSNM to less than 1%. The pull-down NMOS FET is the most vulnerable in the read operation, and the RSNM is degraded by up to 31.68% compared to the pre-irradiation. The RSNM shifts only one VTC due to displacement defects in the single transistor, reducing the closed curve of the butterfly curve by 13.74 mV more than the coupled transistors. As the fin width increases, the RSNM decreases because the gate field is unable to effectively lower the energy barrier raised by displacement defects. When the fin height is reduced, the energy barrier by the gate field is similarly reduced, but the RSNM is reduced because the current per cross-sectional area is larger. The presence of displacement defects in the top of the fin reduces the on-current by up to 31.61% and the RSNM by up to 32.93% due to interference with the main current path. The degradation of both the current and RSNM due to displacement defects saturates as the number of defects in the cluster increases, reaching a point where the energy barrier no longer linearly reduces the carrier mobility. Therefore, FinFET 6T SRAMs with a reduced fin width and increased fin height structure alleviate defect effects.

## Figures and Tables

**Figure 1 micromachines-14-01090-f001:**
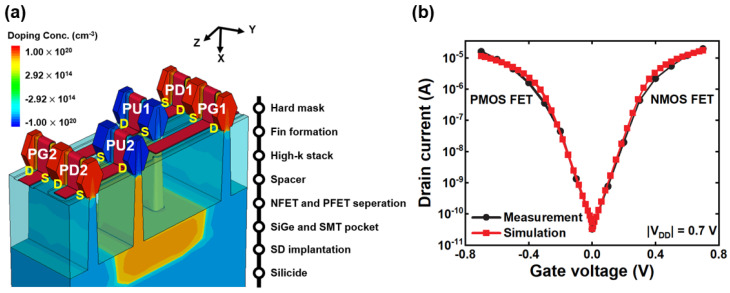
(**a**) Basic structure of simulated FinFET 6T SRAMs; nitride is not shown. (**b**) *I_d_*–*V_g_* transfer characteristics for *V_d_* = *V_g_* = 0.7 V with calibration of experimental measurement [26].

**Figure 2 micromachines-14-01090-f002:**
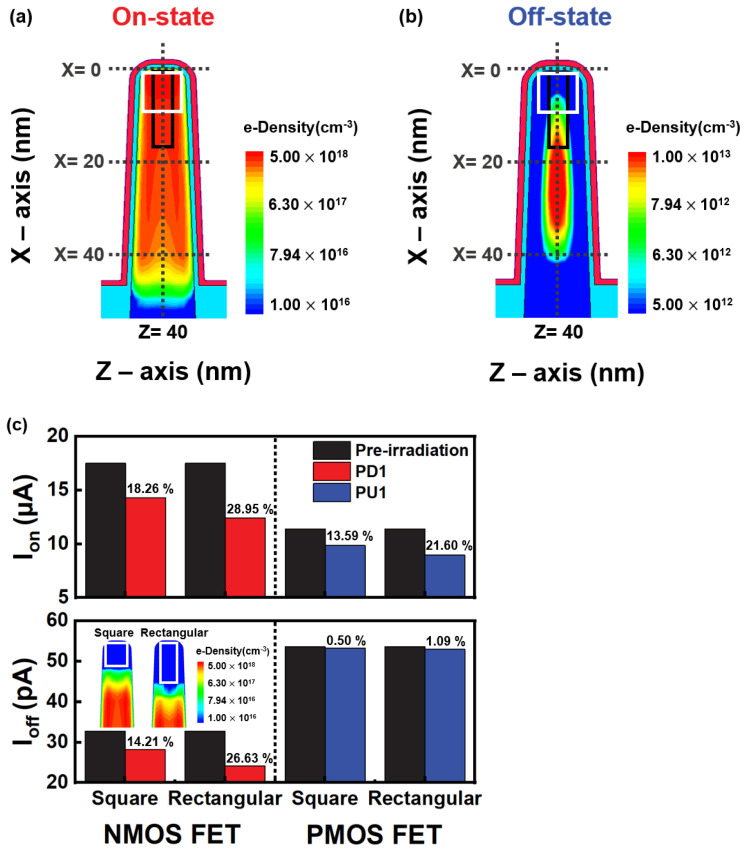
Electron density at (**a**) the on-state and (**b**) off-state. (**c**) Variation in the on- and off-current by the rectangular and square defect clusters.

**Figure 3 micromachines-14-01090-f003:**
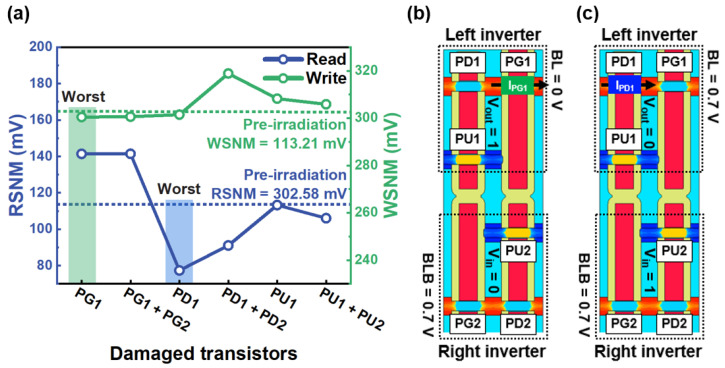
(**a**) Comparison of the read static noise margin (RSNM) and write static noise margin (WSNM) degradation rate using single transistor (PG1 NMOS FET, PD1 NMOS FET, and PU1 PMOS FET) and coupled transistors (PG1 + PG2 NMOS FET, PD1 + PD2 NMOS FET, and PU1 + PU2 PMOS FET) located in the right inverter. (**b**) The write and (**c**) read operation with the fin width and height of 8 and 46 nm, respectively.

**Figure 4 micromachines-14-01090-f004:**
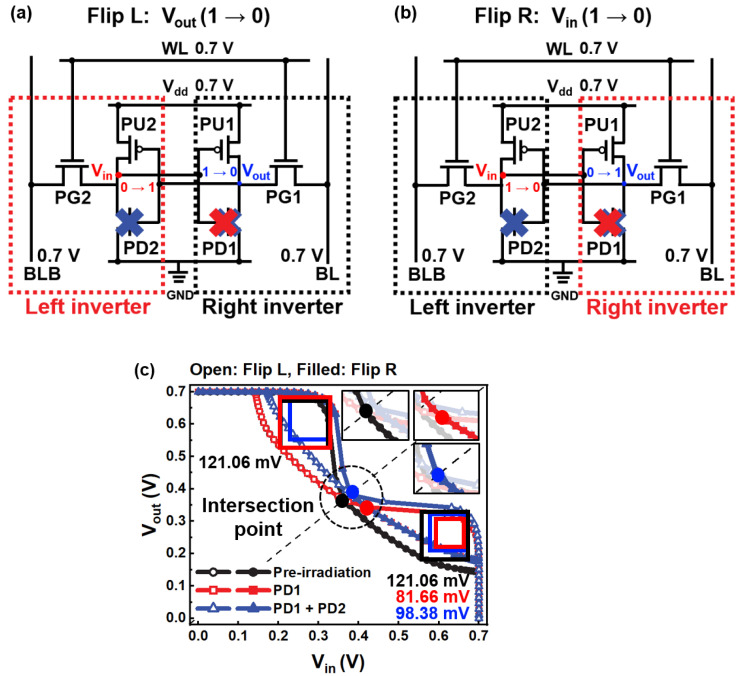
The single transistor (PD1 NMOS FET) and coupled transistors (PD1 + PD2 NMOS FET) where displacement defects were induced during the read operation. (**a**) Flip L (Left VTC), which lowered the output voltage stored as “1” to “0”. (**b**) Flip R (Right VTC), which lowered the input voltage stored as “1” to “0”. (**c**) VTC shift due to displacement defects.

**Figure 5 micromachines-14-01090-f005:**
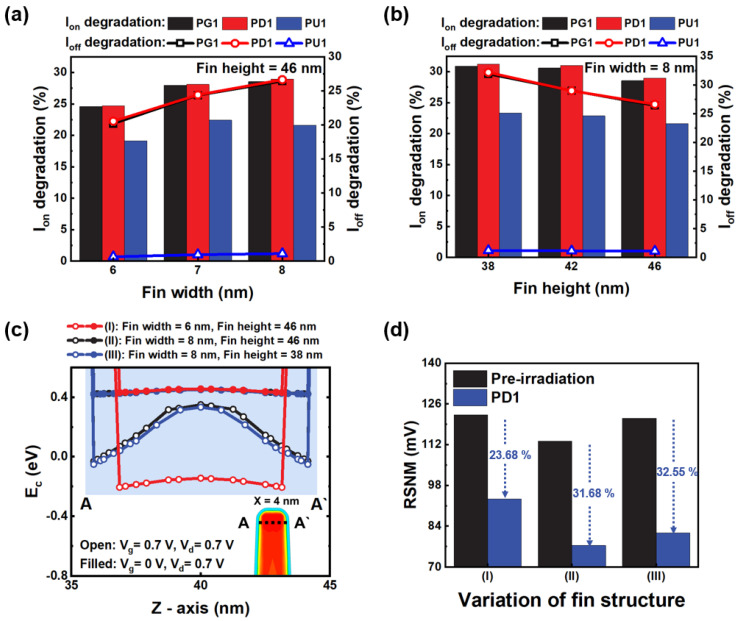
The on- and off-current degradation rates according to variations in (**a**) the fin width (6, 7, 8 nm) and (**b**) fin height (38, 42, 46 nm). (**c**) Conduction band at the on-state (*V_g_* = 0.7 V, *V_d_* = 0.7 V) and off-state (*V_g_* = 0 V, *V_d_* = 0.7 V). (**d**) The RSNM degradation rate of the single transistor and coupled transistors compared by variation in fin structure.

**Figure 6 micromachines-14-01090-f006:**
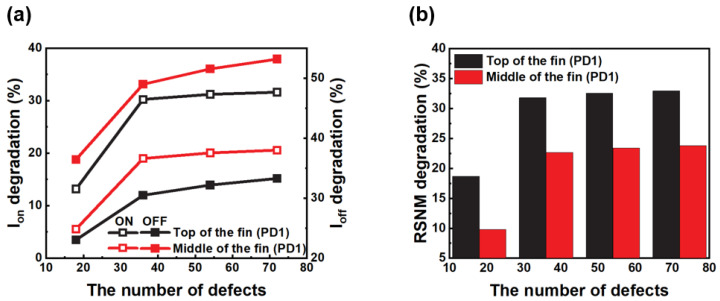
(**a**) The on- and off-current and (**b**) RSNM comparison for defect clusters located at the top and center of the fin.

**Table 1 micromachines-14-01090-t001:** FinFET 6T SRAM geometry and doping concentration.

Parameters	Description	Value
*L_g_*	Gate length	20 nm
*L_eff_*	Effective gate length	20 nm
*W_fin_*	Fin width	8 nm
*H_fin_*	Fin height	46 nm
*CPP*	Contact poly pitch	54 nm
*EOT*	Equivalent oxide thickness	1.4 nm
*N_sd_*	Doping concentration of the source and drain	1 × 10^20^ cm^−3^
*N_ch_*	Doping concentration of the channel	5 × 10^16^ cm^−3^

**Table 2 micromachines-14-01090-t002:** Variation in the RSNM and WSNM due to displacement defects induced in the single transistor and coupled transistors.

Transistor	RSNM (mV)	∆RSNM (%)	WSNM (mV)	∆WSNM (%)
Pre-irradiation	0.11321	-	0.30258	-
PG1	0.1414	−24.90%	0.30048	0.69%
PG1 + PG2	0.14144	−24.94%	0.30063	0.64%
PD1	0.07734	31.68%	0.3015	0.36%
PD1 + PD2	0.09108	19.55%	0.31888	−5.39%
PU1	0.11326	−0.04%	0.30826	−1.88%
PU1 + PU2	0.10607	6.31%	0.30589	−1.09%

## Data Availability

We share research data with the authors of all articles published in the journal MDPI.

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
