# Peer review of "Performance Degradation in Static Random Access Memory of 10 nm Node FinFET Owing to Displacement Defects"

_micromachines, 2023, doi:10.3390/mi14051090_

Round 1

Reviewer 1 Report

The authors have presented a FinFET-based SRAM cell design and simulations work analysed for displacement defects. Authors have worked as variables for a few defect cluster conditions and fin structures. The paper needs a few updates as mentioned here:

1.       Author must add methodology or techniques for SRAM cell design in abstract.

2.       The tool specification should clearly be mentioned in methodology.

3.       Author may add one flow chart on adopted methodology.

4.       Here author must mention about fin and bottom width if it is different.

5.       WSNM or RSNM values should be shown in tabular form and must be compared with other existing work.

6.       What about static and dynamic power loss?, short channel effects? at 10nm technology node.

7.       Author may compare the overall proposed FinFET SRAM with other existing FinFET in terms power consumption, delay, Noise margin

8.       Add some application of such proposed FinFET based SRAM cell.

9.       Author may refer few latest work such:

https://doi.org/10.1007/s12633-022-01935-w

https://doi.org/10.1109/TVLSI.2021.3071940

https://doi.org/10.1007/s11277-020-07765-6

https://doi.org/10.1007/s12633-021-01345-4

https://doi.org/10.1007/s10470-021-01938-4

Author Response

The author submit the response letter. Please, find out the attachement.

Reviewer 2 Report

First, the authors need to revise the introduction thoroughly. Much research has been carried out regarding the effect of radiation on semiconductors, and this is barely mentioned in the manuscript. For the sake of clarity, I would encourage the authors to list the different kinds of radiation-induced degradation mechanisms and then deepen into the one considered for this article.

This being said the way the results are presented is really not easy to follow. it is not clear what metric the authors use to address the degradation. they refer to the noise margins, as the voltage that can be applied without a bit flip...which could be related to the well-known single event upsets phenomenon. I would suggest to structure the results section with the addition of sub headings, and to better present the structure under study, as now authors are using the acronyms PU, PD and PG and it is indeed confusing. 

Honestly, the methodology employed is in my opinion, not adequate, or at least not properly justified. The authors consider a cluster of defects (54 defects, a number this reviewer does not understand where comes from) in a rectangular or squared shape (also an assumption not adequately supported) located in the gate of the transistor (what happens with the defects generated elsewhere?).

Why 54 defects? Authors should consider interacting particles of varying energy and different fluences of irradiation. And if selecting one specific fluence, they should indicate why have they considered that scenario.

Also, it would be greatly appreciated if authors could present a more comprehensive calibration of the simulated device based on the measured electrical characteristics (gate lakeage current vs gate voltage, drain-source current vs gate voltage, capacitance vs voltage, etc.).

English quality is OK. In fact the lack of readability of the manuscript is not due to the English quality but how the text is arranged and organized.

Author Response

(The authors gave the same response as above.)

Round 2

Reviewer 2 Report

Authors have answered my concerns.

Authors have answered my concerns.

Author Response

(The authors gave the same response as above.)
